# AUTOVERSE: AN EVOLVABLE GAME LANGUAGE FOR LEARNING ROBUST EMBODIED AGENTS

## ABSTRACT

We introduce *Autoverse*, an evolvable, domain-specific language for single-player 2D grid-based games, and demonstrate its use as a scalable training ground for Open-Ended Learning (OEL) algorithms. *Autoverse* uses cellular-automaton-like rewrite rules to describe game mechanics, allowing it to express various game environments (e.g. mazes, dungeons, sokoban puzzles) that are popular testbeds for Reinforcement Learning (RL) agents. Each rewrite rule can be expressed as a series of simple convolutions, allowing for environments to be parallelized on the GPU, thereby drastically accelerating RL training. Using *Autoverse*, we propose jump-starting open-ended learning by imitation learning from search. In such an approach, we first evolve *Autoverse* environments (their rules and initial map topology) to maximize the number of iterations required by greedy tree search to discover a new best solution, producing a curriculum of increasingly complex environments and playtraces. We then distill these expert playtraces into a neural-network-based policy using imitation learning. Finally, we use the learned policy as a starting point for open-ended RL, where new training environments are continually evolved to maximize the RL player agent's value function error (a proxy for its regret, or the learnability of generated environments), finding that this approach improves the performance and generality of resultant player agents.[1]

## 1 INTRODUCTION

The idea of open-ended learning in virtual environments is to train agents that gradually get more capable and behaviorally complex. This idea comes in many forms, but what unites them all is that there is no fixed objective or set of objectives; rather, the objectives depend in some way on the agent itself and its interaction with the environment and other agents. This is true for early work on competitive coevolution in evolutionary robotics, work on artificial life simulations, and also for more recent work on open-ended learning.

However, we have yet to see any literally open-ended learning take place in these environments. There have been interesting results, but learning generally stops at a rather low capability ceiling. We hypothesize that this is at least partly because of the poverty of the environments, and the associated limitations in the variability of the environments. It has been observed that the complexity of the behavior of a living being, such an ant or a human, is at least partly a function of the complexity and variability of the environment it is situated in. And it stands to reason that even a very capable and motivated agent would not learn much in an empty white room with no toys, nor in a barren gridworld.

A secondary hypothesis of ours is that open-ended learning is hampered by the complexity of "cold-starting" learning policies from rewards in generated environments, as these may have rare rewards that can only be accessed through uncommon action sequences for which the agents have no priors. This hypothesis suggests that at least part of the reason for the success of reinforcement learning in more well-known domains is that designers, wittingly or unwittingly, build in priors and other domain-specific adaptations to their agents.

In this paper we present Autoverse, a new environment for open-ended learning. Autoverse stands out for allowing more complex environment dynamics and much more environmental diversity than

---

[1]Code is available at https://anonymous.4open.science/r/autoverse-F5A5.

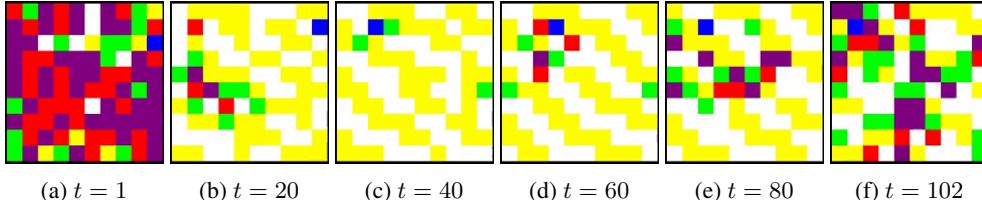

(a) $t = 1$   (b) $t = 20$   (c) $t = 40$   (d) $t = 60$   (e) $t = 80$   (f) $t = 102$

(g) Example of a game which first reaches a relatively stable state (with an oscillating pattern of yellow tile activations), which is then later disrupted by agent actions).

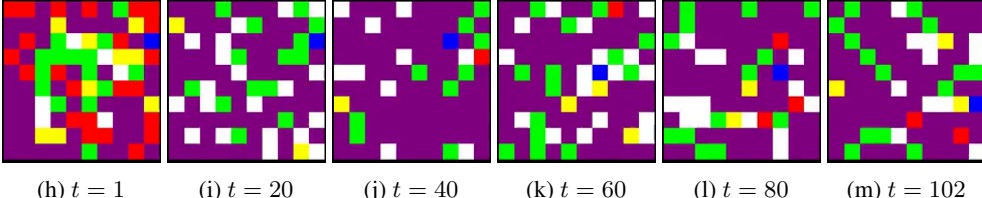

(h) $t = 1$   (i) $t = 20$   (j) $t = 40$   (k) $t = 60$   (l) $t = 80$   (m) $t = 102$

(n) An example of a game which is largely chaotic and unstable, this quality being a common property shared by the majority of evolved games.

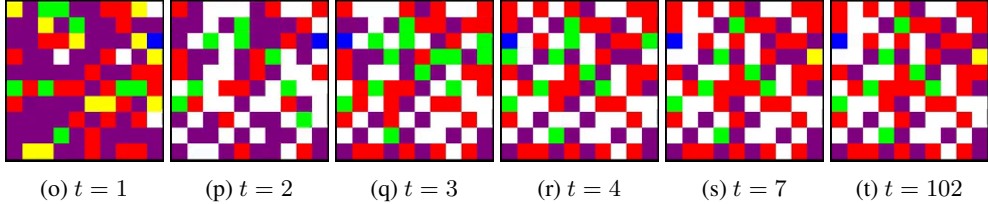

(o) $t = 1$   (p) $t = 2$   (q) $t = 3$   (r) $t = 4$   (s) $t = 7$   (t) $t = 102$

(u) An example of a game in which there is some instability early in the episode, but then reaches a stable state which is maintained for the remainder of the episode.

Figure 1: Examples of environment dynamics in environments evolved for maximum search depth. The player (blue tile) takes the best sequence of actions as returned by a greedy tree search algorithm in order to maximize the reward returned by the environment's transition rules.

other open-ended learning environments. Not only the layout, but almost every aspect of dynamics and interaction can be modified during the open-ended learning process. Environment dynamics are encoded as cellular automata, pairing conceptual simplicity with rich expressivity. The whole system is implemented using JAX, meaning that it run parallelized on GPUs, and at least an order of magnitude speedup.

We also conduct a set of experiments in open-ended learning with Autoverse. In particular, we investigate the value of "warm-starting" reinforcement learning by imitating trajectories taken by best-first tree search agents. This exploits the fact that Autoverse can be used as its own forward model, making rapid tree search practical.

## 2 METHODS

### 2.1 AUTOVERSE: A BATCHED GAME ENGINE WITH EVOLVABLE COMPONENTS

In this section, we develop a framework for batched simulation of grid-world games, allowing game designers to rapidly generate robust agents and complex environments for a broad family of games. We propose a game engine—in the form of a domain specific language (DSL)—that is both general enough to encode a diversity of interesting and complex individual games, while also allowing for batched simulation so as to make rapid agent training accessible on a single GPU. Whereas prior studies have largely fixed the semantics of the generated environments—for example constraining them to always take place in a maze, on 2D navigable terrain (Brockman et al., 2016), or a 2.5D space with moveable objects and rigid-body physics (i.e. XLand Team et al. (2021))—we are interested in generating environments that may carry a broader diversity of possible agent-environment

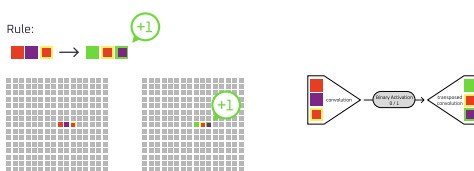

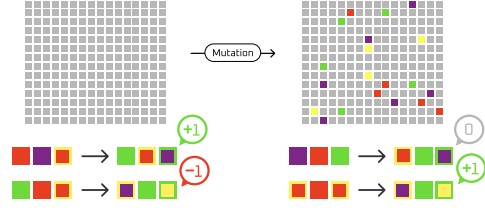

(a) A rule is defined as a sequence of local tile patterns, where the presence of an input pattern causes an output pattern to appear at the following timestep, and the application of a positive or negative reward. A rule set is implemented as a sequence of convolutions.

(b) Environments are mutated by modifying tiles in the initial level map, or in the input/output patterns of rules, as well as the reward values associated with rules.



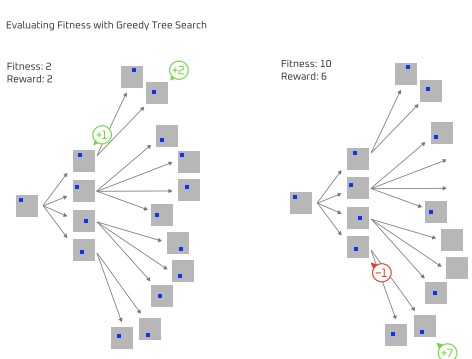

(a) Game environments are selected for maximum fitness—where fitness is defined as the steps-to-best-solution from greedy tree search—and mutated to produce offspring. At each generation, tree search is capped by a maximum number of steps, which is increased when fitness comes within a threshold of this maximum.

(b) For each environment, greedy tree search is performed over the space of possible player actions. The steps taken before finding the best solution is taken as the fitness.

Figure 3: An overview of *autoverse*'s approach to generating novel environments and trajectories.

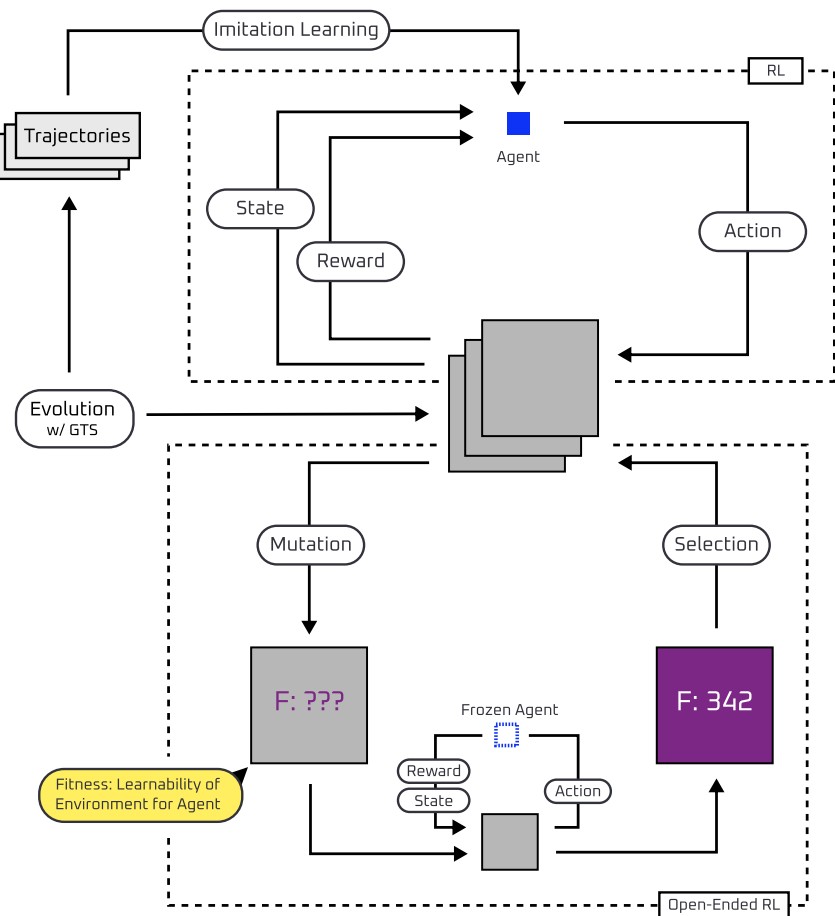

Figure 4: The trajectories and environments generated by *autoverse* are incorporated in an Open-Ended Reinforcement Learning loop.

interactions to further push the generality of OEL-trained controllers. In this section we propose a method to easily batch a surprisingly large category of games.

We focus on games whose dynamics involve discrete elements interacting on a grid. At the core of the DSL are rewrite rules, which specify transformations applied to local patterns of tiles. Despite their seeming simplicity, rewrite rules have been leveraged in prior game description languages, and in particular, in the popular puzzle game engine PuzzleScript (Lavelle, 2013), to generate games ranging from rogue-likes (in which players navigate dungeons, collect treasure and fight enemies), Super Mario Bros-type side-scrolling platformers, and Sokoban-like box-pushing puzzle games and simulacra of circuit-building.

For example, in a roguelike game where a player is tasked with exploring a dungeon littered with obstacles, enemies, and treasure, a rewrite rule might describe the event of a player's stepping into lava by indicating that, if a player tile and a lava tile are overlapping, at the next timestep, the player tile should disappear while the lava remains. A similar logic can be used to allow for basic player movement: we allow the player agent to place invisible 'force' tiles at any cell adjacent to the current player position; we then use a rewrite rule to ensure that whenever a player tile is adjacent to a force tile overlapping with a 'floor' tile (i.e. a grid cell unobstructed by obstacles preventing player movement), at the next timestep, the player should move onto this adjacent floor tile, consuming the force tile in the process.

We propose a novel approach to rewrite rules by taking advantage of the fact that they can be implemented with convolutions, allowing our environment to be both differentiable and easily hardware accelerated. A rewrite rule says that when an $n \times m$ patch $\mathbf{I}$ of tiles is present on the map at timestep $t$, it should be replaced by an $n \times m$ patch $\mathbf{O}$ at timestep $t + 1$. We can express the above statement more formally, focusing on a $n \times m$ patch of the board $\mathbf{C}$ (and supposing patches are one-hot encoded over the number of tile types), taking $\|\mathbf{M}\|_{L_0}$ as the sum of elements in a matrix $\mathbf{M}$ (i.e. $\|\mathbf{M}\|_{L_0} = \mathbf{1}^T \cdot \mathbf{M} \cdot \mathbf{1}$), and letting $I := \|\mathbf{I}\|_{L_0}$, as:

$$\mathbf{B}_{t+1} = \begin{cases} \mathbf{O} & \text{if} \quad \|\mathbf{I} \odot \mathbf{B}_t\|_{L_0} = I \\ \mathbf{B}_t & \text{otherwise} \end{cases}$$

Now, we detail how this operation can be applied to the entire board with a simple sequence of contolutions. First, we construct a convolutional kernel $\mathbf{K}_I$, with dimensions $c \times 1 \times n \times m$ (where $c$ is the number of tile types) for recognizing the input pattern $\mathbf{I}$. To this end, we simply set $\mathbf{K}_I[:, 0, :, :] := \mathbf{I}$, with 0s everywhere else. In this way, when $\mathbf{K}_I$ is applied to a patch containing $\mathbf{I}$, its activations will sum to exactly $I$. We can thereby use this kernel to compute an intermediary binary state $\mathbf{D}$, which is 1 wherever the input pattern is present, and 0 elsewhere. Denoting the state of the board as $\mathbf{C}$,

$$\mathbf{D}_{t+1} = \text{ReLU}\left(\text{conv}_{\mathbf{K}_I}\left(\mathbf{C}_t\right) - I + 1\right)$$

Next, we apply a transposed convolution to generate the change to the board required by corresponding output patterns. More precisely, we construct a transposed convolutional kernel $\mathbf{K}_O$, with dimensions $1 \times c \times n \times m$, such that $\mathbf{K}_O[0, :, :, :] := O - I$. In this way, when $\mathbf{K}_O$ is applied to the binary pattern $\mathbf{D}$, then wherever $\mathbf{D}$ is equal to 1 (and thus the input pattern $\mathbf{I}$ is present at this location), then the result of applying $\mathbf{K}_O$ to $\mathbf{D}$ will, when added back to the prior board state $\mathbf{D}_t$, result in the replacement of the input pattern with the output pattern:

$$\mathbf{C}_{t+1} = \text{conv}_{\mathbf{K}_O}^T\left(\mathbf{D}_{t+1}\right) + \mathbf{D}_t$$

### 2.1.1 POSSIBLE EXTENSIONS TO THE AUTOVERSE LANGUAGE

In addition to binary patterns, we can generate networks for propagating scalar "flows". To simulate a "source" of water using the binary rules above, we might specify that water cells can replicate downward when unobstructed, and otherwise sideways (when unobstructed to the side). When water flows to an adjacent tile, we update an additional channel, denoting the "level" of the water at that point, for example, decrementing once with each horizontal tile transition, such that water is "absorbed" by land tiles after a certain time. Using similar auxiliary, integer-valued channels, we can effectively "count" the distance some substance has travelled from a source, and thereby can move beyond rewrite rules based on local patterns to instantiate more complex algorithms like

breadth/depth-first search-based pathfinding (again as a batched, differentiable convolutional neural networks, as in (Earle et al., 2023)). Though in this work, we limit the games to only involve binary activations, we note that certain games can exhibit phenomena that appear "flow-like", as an emergent property of interaction between evolved rules.

Finally, we note that it is also possible to adapt the rewrite-rules, encoded as convolutions, to support applying each rule only once or a fixed number of times, and/or in a random order, by selecting tiles to rewrite by taking the maximum over an additional channel of ordered or randomly-generated index values. For maximum parallelism, we opt to apply rules in parallel, but can use masking to guarantee that certain rules inhibit others.

## 2.2 Warm-starting open-ended learning from search

### 2.2.1 Evolving game environments to maximize search-based complexity

The first component of our co-learning algorithm involves generating a large and diverse initial set of environment mechanics and layouts prior to agent training. We begin with an effectively empty environment. Here, the player has access to a handful of primitive actions, namely, rotating in either direction, moving forward or backward, and activating a single purple tile at the cell in front of it. In addition to the player, and the purple tiles which it may place, there are 3 additional tile types (rendered in different colors), and an initial random environment is generated by activating one of these additional tile types at each cell on the board with uniform probability. This initial environment contains 5 rules, which begin as no-ops, containing empty input and output patterns of size $3 \times 1$ (all 4 rotated versions of each rules are applied to the board at runtime). During our evolutionary algorithm, these rules may be randomly mutated by changing the value of tiles present in the rewrite rule, and/or by changing the player reward resulting from each application of the rule. The tiles changed may be in the input and/or output pattern, and at various spatial positions relative to one another. For example, an empty rule might eventually be mutated such that it results in a reward of $+1$, contains adjacent purple and red tiles in the input pattern, and a green tile in the output, resulting in a game mechanic wherein whenever the player places a purple tile next to a red tile, it results in a positive reward and mutates the board accordingly. Such a rule would incentivize a play strategy in which the player races to "consume" as many red tiles as possible over the course of an episode. Or, if other rules evolve to result in the propagation of red and/or purple tiles, a more sophisticated and indirect strategy might be preferable. In this way, evolved rules may interact so as to form increasingly complex dynamics, resulting in a lineage of games with non-obvious or perhaps contradictory optimal strategies.

From the standpoint of getting an early sense of an AI player's awareness of and ability to adapt to new rules, the option to mutate reward is a useful feature because it allows for two copies of an environment with identical dynamics to have inverted goals. For example, in one game, red tiles may provide reward, and in another, they may provide negative reward, ensuring that the player cannot ignore the specifics of mutated rules and apply the same strategy to both environments. It is also worth noting that when mutating rewrite rules, we allow for rules to emerge which "kill" the player and end the game (i.e. with one player tile in the input pattern and none in the output), which similarly raises the stakes and decreases the likelihood that a rule-agnostic strategy can be successfully applied to all environments.

We also mutate the initial level layout, a multi-hot array of tiles, by randomly flipping bits in the array. It is important to jointly evolve the initial level layouts as some initial levels, when paired with certain rulesets, may result in unsolvable environments or environments with uninteresting dynamics (e.g. where certain rewrite rules are never applied because some particular tile type necessary for the rule's application is initially absent from the level). Conversely, the same rule-set can result in multiple diverse tasks when paired with different initial level layouts. As a sanity check, we can also disable ruleset-mutation and evolve the initial level layout of any of the base, hand-defined environments, resulting, for example, in mazes or sokoban levels with increasingly difficult solutions (as measured by a search-based agent, described below).

A simple mu + lambda evolution strategy is used to evolve environments using the above-mentioned mutation operators. As a fitness metric, we compute a proxy for the complexity or difficulty of the environment using search. In particular, we use best-first search to explore possible sequences of actions that can be taken by a player agent, prioritizing those trajectories that lead to higher reward.

The fitness of an environment is equal to the number of states visited by search prior to it finding the highest-reward solution. A "node" in the search tree corresponds to a game state, i.e. the current player reward, the position and orientation of the player agent, and the multihot array corresponding to the state of the level at a given timestep; an edge in the search tree is a player action (rotating left or right, moving forward, or activating a tile). When a game state is encountered that is equivalent to some state seen earlier in search, the shallower node—closer to the root of the tree and thereby occurring after fewer player actions—is kept, and the deeper node is pruned from the search tree. If two states are equivalent except for their reward, then only the state with higher reward is kept. The budget of best-first search is limited, and this limit is increased whenever there appears in the population an environment whose best solution required a number of search iterations approaching this limit to some degree.

### 2.2.2 IMITATION LEARNING: DISTILLING SEARCH-BASED SOLUTIONS

Throughout this process of environment evolution, we store trajectories corresponding to the solutions of all environments encountered. If the same environment (i.e. a ruleset and initial level layout) appears twice, we keep the trajectory that led to higher reward. (This situation may arise when a clone of an environment is re-evaluated at a later stage in evolution, with a higher cap on the amount of search afforded to our fitness evaluation.)

We then perform behavior cloning on this archive of trajectories, in effect distilling the set of solutions discovered by search into a neural network. Behavior cloning is a simplistic form of imitation learning, wherein the model is given (state, action) pairs and is trained to predict the corresponding action for each state. Observations consist of a local patch of the surrounding tiles, centered at the player's current position, in addition to a binary representation of the evolved rules of the current environment (so that the agent may adapt its strategy to suit the given mechanics).

### 2.3 OPEN-ENDED REINFORCEMENT LEARNING IN EVOLVING ENVIRONMENTS

Once the behavior cloning algorithm has converged, we continue training the agent with reinforcement learning, randomly sampling at each episode reset from one of the unique evolved environments contained in the set of trajectories above, and using Proximal Policy Optimization (PPO) (Schulman et al., 2017) to update the agent's parameters. Our PPO Jax implementation is based on PureJaxRL Lu et al. (2022) which is in turn adapted from CleanRL Huang et al. (2022), which allows our entire training loop to be just-in-time compiled to run on the GPU. Following work in Unsupervised Environment Design (UED) (Jiang et al., 2021; Parker-Holder et al., 2022), we continue to evolve environments in order to generate an adaptive curriculum for our RL player agent.

Fixing some interval $k_{evo}$ as a hyperparameter, after every $k_{evo}$ updates in our RL loop, we evolve Autoverse environments—both the binary array corresponding to the initial map layout, and the convolutional kernels corresponding to the input-output patterns of the set of rewrite rules. To evaluate the mutated environments, we freeze the weights of the RL-trained player and have it play through 1 or more episodes in the environment. Following Jiang et al. (2021), we compute the mean absolute value function error of the agent over the course of an episode, and use this as the candidate environment's fitness. The value function error is intended as a proxy measure of regret—that is, the difference in return (i.e. discounted reward) accumulated by the learned player over the course of an episode, and that of a hypothetical optimal player. Dennis et al. (2020) show that, when the adversarial loop between the environment-generator agent (in our case an evolutionary algorithm) and player agent is seen as a multi-agent game, wherein the generator's objective to increase, and the player's objective to decrease, such a measure of regret, then this game converges to a Nash equilibrium, implying that the generator has discovered maximally complex and challenging environments with respect to the agent, and the agent has discovered a maximally capable policy with respect to the environments produced by the generator (given some simplifying assumptions). Intuitively, we can think of the value function error as indicating the extent to which the learned agent is "surprised" by the outcome of its episode (i.e. having either over- or under-estimated its performance during play).

Table 1: In agents trained with behavior cloning to imitate the solutions found from greedy search on evolved environments, training and testing performance is higher given larger observations of the surrounding board state. Agents that fully observe the board perform best.

|  | train mean | test mean |
|---|---|---|
| obs window |  |  |
| 5 | 148.23 ± 27.30 | 154.22 ± 12.21 |
| 10 | 124.31 ± 23.53 | 136.69 ± 27.53 |
| 20 | 133.38 ± 15.21 | 145.08 ± 18.95 |
| 31 | **187.87 ± 14.54** | **165.94 ± 16.46** |

Table 2: In agents trained with imitation learning, observing an environments' rules leads to higher performance at train and test time.

|  | train mean | test mean |
|---|---|---|
| observe rules |  |  |
| False | 167.26 ± 12.90 | 151.87 ± 15.49 |
| True | **187.87 ± 14.54** | **165.94 ± 16.46** |

## 3 RESULTS

Tables 1 and 2 show the importance of observations on agent performance when imitation learning on trajectories generated from greedy tree search on evolved environments. Table 1 shows that generally, larger observations of the map allow for increased performance both during training, and on test environments (environments also generated by the evolutionary process, but held out for testing). The best performance comes from agents that are able to fully observe the map (where the observation is centered at the agent's current position, and $0$-padding is added to the map as necessary).

Table 2 shows that agents that are allowed to observe each environment's rule-set perform better than agents for whom the rule-set is replaced by $0$-padding. This shows that the mechanics of the generated environments are sufficiently distinct, such that agents cannot perform effectively without observing the rule-sets.

We show some preliminary qualitative results of the search-based evolutionary process in Figure 1. We observe a variety of distinct environment dynamics in evolved environments. The majority of environments exhibit highly unstable dynamics, in which the majority of cells on the map change state from one timestep to the next, as exhibited in Figure 1n. The prevalence of such environments may partially be explained by the fact that, in an environment where almost all states are different from one another, a search-based agent is less likely to encounter the same state twice, thus forcing it to search longer for optimal states. This is at least true toward the beginning of evolution: all else being equal, if we compare an environment in which the agent's actions have no effect (i.e. there are no rules where the agent can construct the input pattern by placing force tiles) and which is also stable throughout the episode; with an environment in which the agent's actions have no effect, but the map state is changing drastically from one timestep to another, the latter environment will force the agent to construct a larger search tree with more distinct nodes. Given that these chaotic environments remain prevalent later in evolution, however, requires further explanation. It must be the case that these environments are also highly reactive to the actions of the player agent, i.e. that by placing a force tile, the player can put into motion a chain of events (rule-applications) causing a sequence of novel states requiring further search to explore.

One difficulty with the kinds of chaotic dynamics that appear so frequently among evolved environments is the difficulty of interpreting a player-agent's strategy. Future work will qualitatively assess the differences between high/medium/low reward trajectories in such environments.

Another distinct type of environment which we observe in our experiments is exemplified by the evolved environment in Figure 1u. In this environment, there is some activity and state-changes early on during the episode, after which point the map then becomes entirely stable, with the agent

taking no further actions to affect outcomes. Presumably, all of the consequential decisions taken by the player agent occur early on during the episode in this environment. It is surprising, then, that such an environment persists later on in evolution, since one can easily imagine simply extending the complexity of the early-episode phase to later in the episode, thereby increasing potential search complexity. Indeed, it may be that as evolution continues, and the cap on search depth is gradually increased, such exclusively "early-game" environments will die out. On the other hand, it may be the case that the environment is not necessarily restricted to the early game, and that, instead, the vast majority of action trajectories would result in more chaotic behavior. This would be especially understandable if these more chaotic trajectories were deceptively rewarding, with the search agent exploring them in depth before finally arriving at an obscure but long-term rewarding early game move sequence (with this sequence perhaps being one of the rare sequences to result in a stable map state for the rest of the episode). Analysis of alternative action trajectories, along with human testing of the generated environment, can reveal the deeper nature of the evolved dynamics.

Finally, we also observe some environments with something of a balance between relatively fixed/stable states, and more chaotic patterns, as shown in Figure 1g. Here, the environment stabilizes after some initial activity, after which point the agent takes some actions which result in the emergence and propagation across the map of more dynamic structures. Arguably, such environments are the most interpretable: unlike the purely chaotic environments, the impact of the agent's actions are more clearly distinguished against a non-chaotic backdrop; and unlike largely stable environments, the agent's impact on the environment dynamics can be observed over time, instead of occurring in the flash of an instant in a handful of early steps. Another way of seeing this is that it seems less like the agent is learning a fine-grained, carefully-timed and exacting sequence of actions—a kind of rhythmic password—and more like it is planning on a larger scale. Further work would be needed, however, to formalize and quantify the difference between such types of strategies before we might begin to associate them with different environments; ultimately, such heuristics could be used to guide the evolutionary process itself toward environments begetting this type of behavior from player agents. Similarly, our notion of this class of environments being more "interpretable" than those previously described could be pursued further, with the aim of better aligning the open-ended learning process with notions of human interestingness.

## 4 RELATED WORK

Reinforcement learning research has long relied on benchmarks of various kinds. These are often taken from, or inspired by, games, including board games and video games, and sometimes from robotics and other disciplines. An issue with these benchmarks is the risk of overfitting. If the benchmark does not have appropriate degree of variability, the RL algorithm will tend to learn a policy that will work only for a particular configuration of a particular environment. For example, when training deep RL methods to play Atari games in the ALE framework, they will typically learn a policy that works for only one game, and only the particular levels of that game, and break if you give the trained policies a new level or even just introduce visual distortions Zhang et al. (2018); Justesen et al. (2018); Cobbe et al. (2019).

One approach to ensuring sufficient diversity is to rely on procedural content generation (PCG), where levels or other aspects of the benchmark are generated algorithmically Risi and Togelius (2020). While the simplest forms of PCG simply consist of randomly changing parameters or moving things around, there are numerous PCG methods building either on various forms of heuristics and optimization Shaker et al. (2016) or machine learning, including deep learning Liu et al. (2021). Clearly, PCG can help to combat overfitting in RL; by training on an infinite stream of freshly generated levels, more general policies can be found Justesen et al. (2018).

However, diversity in the training set is even better if you have the right sort of diversity. One way of achieving this is through competitive coevolution, where agents seek to perform well in environments and environments seek to provide challenge to agents. This idea, originating in biology Dawkins and Krebs (1979), has a long history in evolutionary computation Rosin and Belew (1997); Hillis (1990), and was later re-discovered in the form of adversarial learning Goodfellow et al. (2014). Applied to generating environments for reinforcement learning, this basic idea has taken on different shapes, including generating environments that provide an appropriate level of challenge or that are learnable by the reinforcement learning algorithm Togelius and Schmidhuber

(2008); Dennis et al. (2020); Bontrager and Togelius (2021); Mediratta et al. (2023). The animating spirit behind much of this work, beyond merely combating overfitting, is to enable open-ended learning.

While there are many benchmarks and testbeds for reinforcement learning methods, few existing benchmarks feature meaningful PCG; exceptions include Obstacle Tower Juliani et al. (2019), Coin-Run Cobbe et al. (2019), and Neural MMO Suarez et al. (2024). In comparison to these, Autoverse is an RL benchmark explicitly relying on and aiming to enable open-ended learning, where environment generation is responsive to progress in agent capabilities.

## 5 CONCLUSION

We introduce *autoverse*, a scalable testbed for open-ended learning algorithms, and run some initial experiments exploring the use of an evolutionary strategy to search for *autoverse* environments comprising difficult game environments with respect to a search-based player agent. We formalize the underlying mechanics of *autoverse* by association with cellular automata and the rewrite-rule approach to game logic developed by other popular game languages. We walk through some examples of popular game and reinforcement learning environments, showing how mazes, dungeons, and sokoban puzzles can be implemented with relatively straightforward sets of rewrite rules. We also show how *autoverse* update rules in general can be implemented with a simple series of convolutions, allowing environment simulation to occur on the GPU, making for fast simulation and neural-network learning in particular within the framework.

Our evolutionary search for challenging environments relative to a search-based player discovers a large number of distinct environments, each constituting a potentially novel and interesting task for an RL-based player agent. This evolutionary search also returns a large number of expert trajectories, which can ultimately be used for imitation learning and to jump start RL. Because the cap on search depth is increased incrementally over the course of evolution, we also obtain a curriculum of increasingly expert trajectories and/or increasingly complex environments. Future work will study how this data can be used to jump-start a generalist reinforcement learning game playing agent by pre-training its weights using imitation learning.

Of particular interest in the evolved *autoverse* environments is the degree to which a given environment's dynamics are stable or chaotic. We note that a large number of environments tend toward chaos, and argue that more human-relevant environments can be found in the middle ground of semi-stable environments, where stable or oscillating patterns tend to be reached, but the player agent can intervent to disrupt or alter them to some degree. Future work is needed to investigate quantitative metrics that may be used to guide the search process toward such environments. More broadly, using pre-trained foundation models or humans-in-the-loop could also allow us both to align the process with notions of human interestingness, as well as to introduce additional human-authored complexity into the learning process.

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
