Ishita Mediratta, Minqi Jiang, Jack Parker-Holder, Michael Dennis, Eugene Vinitsky, and Tim Rocktäschel. Stabilizing unsupervised environment design with a learned adversary. In *Conference on Lifelong Learning Agents*, pages 270–291. PMLR, 2023.

Long Ouyang, Jeffrey Wu, Xu Jiang, Diogo Almeida, Carroll Wainwright, Pamela Mishkin, Chong Zhang, Sandhini Agarwal, Katarina Slama, Alex Ray, et al. Training language models to follow instructions with human feedback. *Advances in Neural Information Processing Systems*, 35: 27730–27744, 2022.

Jack Parker-Holder, Minqi Jiang, Michael Dennis, Mikayel Samvelyan, Jakob Foerster, Edward Grefenstette, and Tim Rocktäschel. Evolving curricula with regret-based environment design. In *International Conference on Machine Learning*, pages 17473–17498. PMLR, 2022.

Sebastian Risi and Julian Togelius. Increasing generality in machine learning through procedural content generation. *Nature Machine Intelligence*, 2(8):428–436, 2020.

Christopher D Rosin and Richard K Belew. New methods for competitive coevolution. *Evolutionary computation*, 5(1):1–29, 1997.

John Schulman, Filip Wolski, Prafulla Dhariwal, Alec Radford, and Oleg Klimov. Proximal policy optimization algorithms. *arXiv preprint arXiv:1707.06347*, 2017.

Noor Shaker, Julian Togelius, and Mark J Nelson. Procedural content generation in games. 2016.

Matthew Siper, Ahmed Khalifa, and Julian Togelius. Path of destruction: Learning an iterative level generator using a small dataset. In *2022 IEEE Symposium Series on Computational Intelligence (SSCI)*, pages 337–343. IEEE, 2022.

Joseph Suarez, David Bloomin, Kyoung Whan Choe, Hao Xiang Li, Ryan Sullivan, Nishaanth Kanna, Daniel Scott, Rose Shuman, Herbie Bradley, Louis Castricato, et al. Neural mmo 2.0: A massively multi-task addition to massively multi-agent learning. *Advances in Neural Information Processing Systems*, 36, 2024.

Open Ended Learning Team, Adam Stooke, Anuj Mahajan, Catarina Barros, Charlie Deck, Jakob Bauer, Jakub Sygnowski, Maja Trebacz, Max Jaderberg, Michael Mathieu, et al. Open-ended learning leads to generally capable agents. *arXiv preprint arXiv:2107.12808*, 2021.

Julian Togelius and Jurgen Schmidhuber. An experiment in automatic game design. In *2008 IEEE Symposium On Computational Intelligence and Games*, pages 111–118. IEEE, 2008.

Guanzhi Wang, Yuqi Xie, Yunfan Jiang, Ajay Mandlekar, Chaowei Xiao, Yuke Zhu, Linxi Fan, and Anima Anandkumar. Voyager: An open-ended embodied agent with large language models. *arXiv preprint arXiv:2305.16291*, 2023.

Chiyuan Zhang, Oriol Vinyals, Remi Munos, and Samy Bengio. A study on overfitting in deep reinforcement learning. *arXiv preprint arXiv:1804.06893*, 2018.

Jenny Zhang, Joel Lehman, Kenneth Stanley, and Jeff Clune. Omni: Open-endedness via models of human notions of interestingness. *arXiv preprint arXiv:2306.01711*, 2023.

# A APPENDIX

## A.1 HUMAN-CONSTRAINED OPEN-ENDED LEARNING

In this section, we discuss in detail several potential approaches for constraining such open-ended learning processes as those afforded by Autoverse to remain in or near the space of human-interpretable or human-relevant games and behaviors.

Standard OEL algorithms—including the one applied here—are based on mathematical approximations of learning potential. However, there is no guarantee that the types of environments that are scored as challenging for an artificial player agent would be of interest to a human. As an example, OEL algorithms can be vulnerable to generating *password-guessing games*, in which environments are generated wherein solutions consist of some lengthy and precise series of actions that must be reproduced exactly in sequence to receive a reward (essentially, the level consists of a password that the agent must guess). These levels are inherently difficult to solve but contain no structure that can be learned.

Previous approaches manage to produce interesting and human-interpretable artifacts by heavily relying on strict constraints built into their simplistic environments. For example, the terrain in the bipedal walker environment Parker-Holder et al. (2022) must always be connected and non-overhanging and the agent morphology fixed; and maze environments always involve some arrangement of wall or empty tiles Parker-Holder et al. (2022); Dennis et al. (2020). However, a more general OEL process, in which new types of entities can be introduced, and new mechanics emerge between them, removes some of these constraints. The larger the number of evolvable components, the more time spent generating password guessing games.

In Autoverse, some structure is necessarily present in the rewrite rules themselves: tiles cannot change arbitrarily, as long as the number of rewrite rules is limited. Whether this underlying structure is "meaningful" in some way is another matter. Insofar as we care about training game-playing agents, then this structure is meaningful if it implements a human-interpretable game, such as the ones we implement here. But as the types of games we want to represent increase in complexity, and we (likely) need more rules to implement them, the space of possibly meaningless games grows along with them.

### A.1.1 JUMP-STARTING LEARNING FROM HUMAN DEMONSTRATIONS

One possibility is to jump-start the OEL process with human demonstrations of trajectories of level construction or examples of completed human-authored environments. We can then use imitation learning or other distribution-matching methods such as GANs or diffusion models to initialize controllers and environment generators whose behavior is aligned with these human demonstrations. This allows us to skip an early and long phase of level generation where levels are highly random and uninteresting. Due to the sparse reward that level generators frequently receive, this phase can dominate training time. Prior work has explored imitation learning on reverse trajectories of random destruction to produce human-like environment generators Siper et al. (2022) or aligned with human heuristics for level quality Khalifa et al. (2020); Earle et al. (2021; 2022); Jiang et al. (2022). However, it is not yet clear whether these approaches can accelerate the quality of agents trained in an open-ended learning loop. Future work should investigate whether the incorporation of these imitated levels, possibly as a constraint on statistical distances between the level generator and the imitation policy, will lead to agents that are more robust to hold-out test games authored by human designers.

### A.1.2 USING FOUNDATIONS MODELS TO JUDGE HUMAN-RELEVANCE OF LEVELS

Another option to use large multi-modal models trained on massive amounts of human data to inject useful priors of human-relevance into the learning process. We can assess environments and agents not only for their learnability and capability, respectively, but also for the degree of relevant novelty according to the pre-trained foundation model. Due to their training on a massive corpora of text and video data, these models may have partially internalized notions of what types of levels are realistic or interesting. Such an approach has already seen some success in generating a curriculum of interesting player behaviors in the open-world sandbox game Minecraft, with a Large Language

Model (LLM) being used to generate a curriculum of increasingly novel tasks Wang et al. (2023); Zhang et al. (2023).

**?** show that foundation models have sufficient internal knowledge of level structure to generate in-game structures and environments—again in Minecraft—from text descriptions. This is achieved by extending an existing text-to-3D method, which uses a text-image model to guide the optimization of a NeRF given a text prompt. In particular, the NeRF is quantized so as to produce a discrete representation, which is then mapped to a voxel representation of in-game assets. One can imagine extending this approach of using foundation models as a translator between modalities to perform text-to-rewrite-rule and text-to-agent behavior. This is especially true in Autoverse: since the application of rewrite rules is differentiable, gradients can be backpropagated through episode rollouts to optimize discrete rule representations using the quantization method described above, with gradients corresponding to contrastive loss from a vision-language model that is attempting to match generated environment/agent mechanics/behavior, rendered as a sequence of frames from gameplay, to a text description

An alternative approach, not relying on the differentiablity of the environment, would be to create a small dataset of game mechanics and their corresponding natural language descriptors and provide these as in-context prompts to a foundation model. While past work has demonstrated that this is effective in simple text-domains Kwon et al. (2023) to create particular agent characteristics, whether the in-context-learning capabilities of foundation models extend to more complex domains has not been studied.

### A.1.3 OEL WITH HUMAN FEEDBACK

Several forays have been made into the space of human-in-the-loop environment generation, and these could be extended to incorporate learned player agents. Charity and Togelius (2022) published a website in which users were invited to design environments given one of a collection of simple 2D tile-sets. A bot would then publish polls on Twitter, with images of two random environments, asking users to vote for that which was more aesthetically pleasing. This human preference data was used to continually train a discriminator to predict the quality of environments, which in turn was used to guide the training of an environment generator, whose output was then included in the set of environments to be judged by human users on Twitter. In Charity et al. (2020; 2022), a Quality Diversity algorithm is used to search for playable levels in the popular indie box-pushing puzzle game *Baba is You*, with an emphasis on environments that require engagement with a diversity of game mechanics (in the game, solving a level usually involves altering one or several of its mechanics, e.g. temporarily giving the player control over a rock instead of the "Baba" avatar, by pushing into place a row of tiles reading "rock is you"). These prior works demonstrate that this seemingly laborious process of having a human-in-the-loop is actually scalable given crowdsourcing.

However, these works only focus on the interesting components of the level that are immediately available from a static depiction of the level and do not account for the behavior of an agent in the level e.g. is this level interesting to play, does the agent's behavior correspond to being stuck, would this have been a level where they (the human) would have learned a new skill or idea, etc. We propose to generate sets of paired level play-throughs and ask users to rate which playthrough they prefer (optionally neither) which can then be used to train a reward model to score levels. After enough manual human feedback, a surrogate model of human-relevant novelty could be trained, helping the OEL process to continue generating relevant novelty without the need for constant, time- and labor-intensive human feedback. (We can think of this as Open-Ended Learning through Human Feedback, by analogy with the RLHF technique used to guide pre-trained LLMs toward helpful outputs Ouyang et al. (2022).)

### A.2 IN-DEPTH EXAMPLES

In this section, we formally express implementations of a few canonical games in terms of Autoverse's rewrite rules.

Here, we express an environment in terms of a set of rewrite rules, with the $i^{th}$ rewrite rule being given in terms of its input and output patterns $\mathbf{I}_i, \mathbf{O}_i$. Each of these tensors is a $c \times n \times m$-size array (where $c$ is the number of tile types in the environment, and $n \times m$ is the size of a 2D patch on the board) with values in $\{-1, 0, +1\}$. In particular, we define these tensors by populating them with

size-$c$ onehot vectors used to identify the presence of particular tile types (denoted TILE to, e.g., indicate the onehot vector that would correspond to the presence of a tile of type "Tile" on the map). The reward resulting from the application of the $i^{th}$ rule is denoted $R_i$.

### A.2.1 MAZE

To express a simple maze environment, where the player can traverse empty (but not wall) tiles, and is rewarded when it consumes a goal/food tile, we restrict player action to place a FORCE tile at any tile adjacent to the player's position, and restrict map generation/mutation such WALL and EMPTY tiles cannot overlap in the initial map layout. Then, player movement is given by

$$\mathbf{I}_0 = \left[\left[ \text{ PLAYER,} \quad \begin{matrix} \text{FORCE+} \\ \text{EMPTY} \end{matrix} \right]\right] \quad \mathbf{O}_0 = \left[\left[ \quad \mathbf{0} \quad , \quad \begin{matrix} \text{PLAYER+} \\ \text{EMPTY} \end{matrix} \right]\right]$$

and food-consumption is given by

$$\mathbf{I}_1 = \left[\left[ \text{ PLAYER,} \quad \begin{matrix} \text{FORCE+} \\ \text{FOOD} \end{matrix} \right]\right] \quad \mathbf{O}_1 = \left[\left[ \quad \mathbf{0} \quad , \quad \begin{matrix} \text{PLAYER+} \\ \text{EMPTY} \end{matrix} \right]\right] \quad R_1 = 1$$

Note that $\mathbf{I}_0, \mathbf{O}_0, \mathbf{I}_1, \mathbf{O}_1$ all have dimensions $c \times 1 \times 2$. Since, by default, rules are broken down into all possible rotations, these rotated patches will capture all $4$ possible adjacent arrangements of PLAYER and FORCE+EMPTY, and PLAYER and FORCE+FOOD activations, respectively. Also recall that the projection kernel $\mathbf{K}_O$, which serves to project the output pattern into the board state at the following timestep (via transposed convolution) is populated with $\mathbf{O} - \mathbf{I}$. So, the PLAYER activation at cell $(0, 0)$ in $\mathbf{I}_0$ will be removed when the rule is applied, even though $\mathbf{O}_0[:, 0, 0] = \mathbf{0}$.

If we wanted to additionally penalize the player for attempting to traverse a wall tile, we could add the rule:

$$\mathbf{I}_2 = \left[\left[ \text{ PLAYER,} \quad \begin{matrix} \text{FORCE+} \\ \text{WALL} \end{matrix} \right]\right] \quad \mathbf{O}_2 = [[ \text{ PLAYER, } \text{ WALL }]] \quad R_2 = -1$$

Or, if in such an event we wanted to penalize the player and also kill them, ending the episode, essentially treating the wall like lava, we could instead write:

$$\mathbf{I}_2 = \left[\left[ \text{ PLAYER,} \quad \begin{matrix} \text{FORCE+} \\ \text{WALL} \end{matrix} \right]\right] \quad \mathbf{O}_2 = [[ \quad \mathbf{0} \quad , \text{ WALL }]] \quad R_2 = -1$$

### A.2.2 SOKOBAN

To implement Sokoban, we once again restrict player actions to involve the placement of FORCE at adjacent tiles, and restrict WALL activations from overlapping with EMPTY, CRATE, and TARGET activations in the initial map layout. Rule $0$ involving player movement onto empty tiles is identical to that from the maze environment above. To allow the player to push crates onto empty tiles, we write:

$$\mathbf{I}_1 = \left[\left[ \text{ PLAYER,} \quad \begin{matrix} \text{FORCE+} \\ \text{CRATE} \end{matrix} , \text{ EMPTY} \right]\right] \quad \mathbf{O}_1 = \left[\left[ \quad \mathbf{0} \quad , \quad \begin{matrix} \text{PLAYER+} \\ \text{EMPTY} \end{matrix} , \text{ CRATE} \right]\right]$$

To provide reward at each step for all crates that are positioned on targets, we write:

$$\mathbf{I}_2 = \left[\left[ \begin{matrix} \text{CRATE+} \\ \text{TARGET} \end{matrix} \right]\right] \quad \mathbf{O}_2 = \left[\left[ \begin{matrix} \text{CRATE+} \\ \text{TARGET} \end{matrix} \right]\right] \quad R_2 = +1$$

### A.2.3 BOULDER DASH

Taking Boulder Dash as an example, we can sketch how to implement the game mechanics using rewrite rules. The player controller has access to 4 actions, each of which will activate the "force" channel at one of the 4 tiles adjacent to the player avatar. A force tile that overlaps with one of the tiles constituting the hard border of the game map will disappear (corresponding to an invalid move action). A force tile that overlaps with a dirt tile will cause a rewrite that moves the player into the dirt tile, while removing the dirt and force activations. A force tile applied to a boulder, where the next tile over is empty, results in the boulder moving into this empty tile, and the player into this formerly boulder-occupied tile. Wherever a boulder is above an empty tile, the boulder will descend into the empty tile, and when a boulder is above a wall or solid tile that has an empty tile to its left or right, the boulder will descend diagonally into this empty tile. When the player overlaps with the goal tile, they will be granted a reward of 1.

Formally, digging is given by the rule,

$$I_0 = \left[\left[ \text{PLAYER}, \begin{array}{c} \text{FORCE+} \\ \text{DIRT} \end{array} \right]\right] \qquad O_0 = \left[\left[ \mathbf{0} \;,\; \begin{array}{c} \text{PLAYER+} \\ \text{EMPTY} \end{array} \right]\right]$$

boulder-pushing by

$$I_1 = \left[\left[ \text{PLAYER}, \begin{array}{c} \text{FORCE+} \\ \text{BOULDER} \end{array} ,\; \text{EMPTY} \right]\right] \quad O_1 = \left[\left[ \mathbf{0}, \begin{array}{c} \text{PLAYER+} \\ \text{EMPTY} \end{array} ,\; \text{BOULDER} \right]\right]$$

and boulder falling by

$$I_2 = \left[ \begin{array}{c} [\text{BOULDER}], \\ [\text{EMPTY}] \end{array} \right] \qquad O_2 = \left[ \begin{array}{c} [\text{EMPTY}], \\ [\text{BOULDER}] \end{array} \right]$$

(without rotations), and boulder rolling-and-falling by

$$I_3 = \left[ \begin{array}{cc} [\; \text{BOULDER}, & \text{EMPTY} \;], \\ [\; \text{-EMPTY}, & \text{EMPTY} \;] \end{array} \right] \qquad O_3 = \left[ \begin{array}{cc} [\; \text{EMPTY}, & \text{EMPTY} \;], \\ [\; \text{-EMPTY}, & \text{BOULDER} \;] \end{array} \right]$$

(with flipping along the vertical axis; also note that $-\text{EMPTY}$ in the bottom-left of the output pattern means only to leave this cell unchanged, given that the corresponding position in the decoder kernel $K_1$ will be $-\text{EMPTY} + \text{EMPTY} = 0$).