# OpenReview forum: "Autoverse: an Evolvable Game Language for Learning Robust Embodied Agents"
_ICLR.cc/2025/Conference — Submitted to ICLR 2025_

### Official Review · Reviewer_Y6Rs · 2024-10-22

**Soundness:** 2
**Presentation:** 1
**Contribution:** 3
**Rating:** 5
**Confidence:** 3

**Summary:**

The paper introduces Autoverse, a domain-specific language (DSL) for creating 2D grid-based games, designed to enhance open-ended learning for Reinforcement Learning (RL) agents. Autoverse allows for evolving complex game environments using cellular-automaton-like rewrite rules, which can be parallelized on the GPU to speed up RL training. The authors propose a framework involving the evolution of environments, imitation learning from search-based solutions, and reinforcement learning in evolving environments to generate increasingly challenging tasks for RL agents. This approach aims to overcome the challenges of cold-starting RL in open-ended environments and produce more behaviorally complex and adaptable agents. They first evolve Autoverse environments (their rules and initial map topology) to maximize the number of iterations required by greedy tree search to discover a new best solution, producing a curriculum of increasingly complex environments and playtraces. The proposed method then distill these expert playtraces into a neuralnetwork-based policy using imitation learning. Finally, the learned policy becomes as a starting point for open-ended RL, where new training environments are continually evolved to maximize the RL player agent’s value function error (a proxy for its regret, or the learnability of generated environments), finding that this approach improves the performance and generality of resultant player agents.

**Strengths:**

1, Originality: The paper presents Autoverse, a new environment for open-ended learning, which allows more complex environment dynamics and much more environmental diversity than other open-ended learning environments.
2, Quality: The use of JAX for implementing cellular-automaton rules allows efficient parallelization on GPUs, and at least an order of magnitude speedup.
3, Significance: The use of imitation learning followed by reinforcement learning provides a structured way for agents to learn from expert play traces and then further refine their behavior, leading to better generalization.

**Weaknesses:**

1, Presentation Improvement: Provide more detailed feedback, such as clarifying ambiguous elements like the meaning of the gray box in Figure 4 and addressing layout issues with full-page figures such as figure 3 and figure 4. And the texts in figure 3 are too small.
2, Scalability Evidence: Request empirical evidence or additional explanations to demonstrate scalability, including experiments or metrics.
3, Justification for Greedy Tree Search: Maybe adding references in related work and experimental validation to justify the choice of greedy tree search.
4, Comparison of Methods: Recommend including a comparison between the proposed approach and using only imitation learning or reinforcement learning.
5, Performance Results: The paper mentioned "Once the behavior cloning algorithm has converged, we continue training the agent with reinforcement learning", but there is no results show that what kind of performance can this method reach when behavior cloning algorithm has converged and what is the final performance compared to that stage.
6, Lack explanation of comparing with other environments: In the paper "Autoverse stands out for allowing more complex environment dynamics and much more environmental diversity than other open-ended learning environments." But I did not see any citation or detailed comparison with other open-ended learning environments. Further discussion will make the arguments more convincible.

**Questions:**

In general, I would really appreciate it if the authors could provide more details and discussions of the proposed method performance and the reasons why they design the whole method and environment in this way. And also, the comparison with other baseline or ablation study will be welcomed.

---

### Official Review · Reviewer_e4bs · 2024-10-28

**Soundness:** 2
**Presentation:** 2
**Contribution:** 3
**Rating:** 3
**Confidence:** 3

**Summary:**

The paper introduces Autoverse, a special programming language for creating different types of grid-based game environments that help with open-ended learning in RL. Autoverse lets people build complex, changing game settings, especially for 2D, single-player games like mazes, Sokoban puzzles, and dungeon exploring. It uses simple rules, based on cellular automata, to quickly create these environments, which run well on a GPU. The system first teaches RL agents by having them imitate expert examples, then moves to open-ended RL in constantly changing environments to make the agents more adaptable.

**Strengths:**

Autoverse offers efficiency by leveraging GPU-based batch processing. Its rewrite rule framework enables the creation of a vast array of dynamic, grid-based game environments, enhancing agent adaptability and preventing overfitting to static setups. The method's progressive curriculum, which evolves increasingly complex environments, allows agents to improve incrementally, while the integration of imitation learning with RL provides agents with a solid starting foundation. These strengths make Autoverse an innovative and robust tool for advancing research in open-ended learning.

**Weaknesses:**

First, there are presentation issues, as figures (like 3 and 4) lack clarity—small text sizes, ambiguous elements, and layout issues reduce readability. Second, empirical evidence demonstrating scalability is lacking; additional experiments or metrics require to solidify this claims.  The paper fail to do performance comparisons of Autoverse’s combined imitation and RL approach against using either method alone. While it states that Autoverse’s environments are more complex and diverse than others, this claim lacks citations or direct comparisons with other open-ended learning benchmarks. Finally, the paper does not report the performance achieved after the behavior cloning stage, leaving its contribution to the final results ambiguous. Addressing these points could greatly enhance the paper’s clarity, rigor, and persuasive power.

**Questions:**

1. Please clarify any confusing parts in the figures and tables.

2. Please provide additional experiments to support each claim in the paper, such as scalability and comparisons with other environments.

3. Please ensure that the figures and results are presented neatly and clearly.

---

### Official Review · Reviewer_a3dn · 2024-10-29

**Soundness:** 1
**Presentation:** 1
**Contribution:** 2
**Rating:** 3
**Confidence:** 4

**Summary:**

The paper introduces a novel 2D and grid-like environment framework, named Autoverse, based on Cellular Automata (CA).  They define environments using the initial state of the CA, and the update rules (including state transition rules and reward). This allows authors to evolve environments in an open-ended RL setting. Moreover, the authors also propose an open-ended RL method based on bootstrapping the open-ended learning by pre-training the agent using an initial set of evolved environments and expert trajectories from a search method. Experiments show that the evolution process generates many chaotic environments, stable environments, and others in between (the ones of most interest). Furthermore, the open-ended method seems to benefit from fully observing the environment and access to the update rules that update the cells of the CA.

**Strengths:**

- The idea of using CA-based environments is both original and interesting. The fact that practically limitless environments can be created by modifying the initial state and the update rules is very interesting from an open-ended learning perspective.

- I also find CA environments relevant for research on foundation models for RL. This type of environment could be very valuable for generating training data for these types of models.

- I think that the presentation of Autoverse (Section 2.1) is clear and can be easily understood, whereas Figure 1 also helps to visualize the types of environments that are generated by the evolution process.

**Weaknesses:**

Although I think that the main idea (evolvable CA environments) is very interesting and could be promising for areas as foundation models for RL, UED, and open-ended learning. I have important concerns on several aspects of the paper:

- Although most ideas are clearly explained, the overall presentation and soundness of the paper are poor.
    + The introduction section makes many (non-trivial) statements with no references. The introduction has no references, which I found surprising. Some examples of such sentences missing references and evidence/support are:
        - L31: "The idea of open-ended learning in virtual environments is [...]"
        - L32: "This idea comes in many forms, but what unites them all is that [...]"
       - L38:  "There have been interesting results, but learning generally stops at a rather low capability ceiling."
       - L40: "It has been observed that the complexity of the behavior of a living being, [...]"

- The ability to endlessly evolve and generate new environments is compelling, but as mentioned multiple times in the paper (e.g.,   L71, L413) evolutive process generates very unstable environments in most cases. I have serious concerns about the ratio of unusable and usable environments generated by the evolutive process. Authors claim scalability, but how much computational resources are needed to generate a fair amount of actually usable scenarios? I think that an exhaustive analysis of this topic is crucial and missing in the current version of the paper.

- I think that the paper misses many experimentation details. For instance, experiments shown in Table1 and 2 are missing information on the number of evaluations, the number of generated test/train environments, the number of repetitions, etc. Seems that this information is also missing in the appendix. Moreover, none of the appendix sections are referenced in the main paper. Please, consider providing as many details as possible on the experimentation to ensure transparency and reproducibility. Moreover, I strongly believe that all sections of the appendix should be referenced at least once in the main paper.

-   I strongly believe that experimentation should be improved. The authors propose a "warm-start" method for open-ended RL, but they do not provide any evidence for why this method is relevant for the research community. I won't ask for a comparison with tens of methods from previous literature, but proposing a new method at least requires an exhaustive analysis of it. Some examples of how I think this could be improved:
    - Reward values in Tables 1 and 2 are missing interpretability. I can see that the results of some settings are better than others, but how much? Having some sort of reference value could help (e.g., results of a random agent or some well-known baselines).
    - Why is this method interesting? Does it obtain better results? Does warm-staring help to improve the results? Many experiments could be presented to answer these questions.

- The paper claims computational efficiency but no evidence is provided. Experiments/benchmarks on computational performance are missing if such claims are made.

- I think that the presentation has room for improvement. Some examples:
     - Figure 1 is presented is located in page 2, but is referenced on page 8 for the first time.
     - Figure 3 holds an entire page but some of its text (Fig 3.a) is **extremely** small. There is plenty of blank space on Page 3 to increase some parts of the figure or rearrange them to improve visibility.
    - The format of tables 1 and 2 is not aligned with the ICLR style. Please carefully read the style PDF before submitting the paper.
    - There are many incorrect usages of parenthesis in references. For example, parenthesis in Earle et al. 2023 L271 should be removed, and Section 4 is missing parentheses in most of its references: L470-471, L476, L482...

**Questions:**

- Being able to generate a vast number of environments is very interesting from an open-endedness perspective. However, many of the features that enable such versatility and efficiency limit Autoverse to grid-like, 2D, and seemingly small environments. Can methods developed in Autoverse be applied to more realistic scenarios (that are ultimately of interest)? Examples of these alternative (also open-ended) scenarios are Craftax  (Matthews et al., 2024)  or MineDojo (Fan et al., 2022). If not applicable, then what is the relevance of Autoverse?

- Is the grid size constant in all of the generated environments? If not, would it be possible to modify the framework to hold grids of different sizes?

- How does the search method (used for the "warm-start") scale with the size of the grid and number of actions?

- What is (approximately) the ratio of unusable and usable environments generated by the evolutive algorithm? With unusable I refer to environments so unstable that are not beneficial for the optimization process of the RL agent.

- Do the reward update rules always take into account the actions of the agent? or reward can be generated by multiple cells of the CA interacting with each other with no relation to the actions of the agent?

---

### Official Review · Reviewer_t3u1 · 2024-11-12

**Soundness:** 1
**Presentation:** 1
**Contribution:** 2
**Rating:** 3
**Confidence:** 4

**Summary:**

This paper proposes Autoverse, an evolvable domain-specific language for single-player 2D grid-based games, as a training ground for Open-Ended Learning (OEL) algorithms. It describes game mechanics through rewrite rules similar to cellular automata, combines evolutionary algorithms to generate complex environments, employs imitation learning and reinforcement learning to train agents, and experimentally studies the impact of the observation range on agent performance and the dynamic characteristics of the evolved environments.

**Strengths:**

- The design of Autoverse combines a domain-specific language with cellular-automaton-like rewrite rules and achieves efficient computation through convolutions, providing new perspectives and methods for game environment generation and agent training.
- Through the evolution of the environment, the complexity of the environment can be gradually increased according to the search ability of the agent, effectively avoiding the agent from prematurely falling into local optimal solutions and also providing a curriculum learning from simple to complex for the agent.

**Weaknesses:**

1. Overall, the paper has limitations in terms of innovation. The framework of combining imitation learning and reinforcement learning used is not novel and has been involved in many current related research fields.
2. Although the rewrite rules are highly expressive, they are difficult to understand and interpret, which may limit their promotion and further development in practical applications, especially when manual intervention of the rules is required.
3. In the process of environment evolution, only mutation operations are involved, and crossover operations are not included. This may limit the exploration range of the diversity of environment generation to some extent.
4. The main experiment is lacking. There is a lack of direct comparison experiments with other advanced methods, making it difficult to accurately evaluate the advantages and disadvantages of Autoverse and clearly define its competitiveness in this field.
5. The long-term training effect experiment is insufficient. The experimental results mainly present the immediate performance data under specific settings. The performance change trend after a significant increase in the training cycle, the stability of the agent's strategy after a large number of environmental changes, and the evolution law of the long-term adaptability to new environments are not provided, making it difficult to judge the long-term comprehensive ability development.
6. The coherence of the chapter structure is poor in some parts. For example, the transition from the method description to the experimental results is not natural, affecting the reader's understanding of the logical relationship of the paper.

**Questions:**

1. How can the diversity of the generated environments be evaluated more accurately and quantitatively? Besides the current qualitative analysis, are there other more objective and comprehensive indicators to measure the differences and complexity between different environments to better prove the effectiveness of environment generation?
2. How does the paper dynamically regulate the evolution speed of the environment to avoid being too fast for the agent to learn or too slow to cause resource waste?

---

### Meta-Review · Area_Chair_3jrr · 2024-12-19

**Metareview:**

Autoverse is a highly-original concept for a new open-ended learning environment. The ideas and engineering behind this work are creative. However, as all reviewers unanimously agreed, the presentation and experimental design have much room for improvement. Unfortunately, the current gap between the environment concept and the presentation of the empirical component of this work makes it unfit for publication at a top venue like ICLR.

Specifically, the authors should seek to incorporate clearer quantitative comparisons that highlight the strengths of Autoverse as a benchmark environment for open-ended learning with respect to previous environments. For example, more in-depth analysis of the evolutionary dynamics during training and a clear quantitative comparison demonstrating a) how Autoverse environments are more diverse than previous environments, and b) how Autoverse environments provide a greater challenge to existing methods and therefore provides signal that previous benchmarks do not, which is at a high level, a central motivation for Autoverse.

**Additional Comments On Reviewer Discussion:**

Neither authors nor reviewers did not engage during the rebuttal period.

---

### Decision · Program_Chairs · 2025-01-22

Reject